# Reduction of ZTD outliers through improved GNSS data processing and screening strategies

Katarzyna Stepniak[1], Olivier Bock[2], Pawel Wielgosz[1]

[1] University of Warmia and Mazury in Olsztyn, Olsztyn, 10-719, Poland
[2] IGN LAREG, University Paris Diderot, Sorbonne Paris Cité, 75013 Paris, France

*Correspondence to*: Katarzyna Stepniak (katarzyna.stepniak@uwm.edu.pl)

**Abstract.** Though Global Navigation Satellite System (GNSS) data processing has been significantly improved over years it is still commonly observed that Zenith Tropospheric Delay (ZTD) estimates contain many outliers which are detrimental to meteorological and climatological applications. In this paper, we show that ZTD outliers in double difference processing are most of the time caused by sub-daily data gaps at reference stations which cause disconnections of clusters of stations from the reference network and common–mode biases due to the strong correlation between stations in short baselines. They can reach a few centimetres in ZTD and coincide usually with a jump in formal errors. The magnitude and sign of these biases are impossible to predict because they depend on different errors in the observations and on the geometry of the baselines. We elaborate and test a new baseline strategy which solves this problem and significantly reduces the number of outliers compared to the standard strategy commonly used for positioning (e.g. determination of national reference frame) in which the pre-defined network is composed of a skeleton of reference stations to which secondary stations are connected in a star-like structure. The new strategy is also shown to perform better than the widely-used strategy maximising the number of observations which available in many GNSS software. The reason is that observations are maximised before processing whereas the final number of used observations can be dramatically lower because of data rejection (screening) during the processing. The study relies on the analysis of one year of GPS (Global Positioning System) data from a regional network of 136 GNSS stations processed using Bernese GNSS Software v.5.2. A post-processing screening procedure is also proposed to detect and remove a few outliers which may still remain due to short data gaps. It is based on a combination of range checks and outlier checks of ZTD and formal errors. The accuracy of the final screened GPS ZTD estimates is assessed by comparison to ERA-Interim reanalysis.

## 1 Introduction & motivation

Outliers and gaps in ZTD (Zenith Total Delay) series estimated from ground-based GNSS data are detrimental to meteorology and climate monitoring applications (Bock et al., 2016). Though GNSS data processing has been significantly improved over years, outliers are still frequently observed in ZTD time series. This study aims at understanding the main factors leading to these outliers and testing improved processing strategies capable to minimize these effects in the context of post-processing of

GNSS data from moderate-size networks (e.g. national scale). We show that the baseline design strategy in a double-difference network processing has a strong impact on the quality and continuity of ZTD time series.

Another approach of satellite data processing which can be used to estimate GPS ZTD is the precise point positioning (PPP) technique. Since PPP allows to process each station individually, there is no direct propagation of errors between stations. However, the accuracy of ZTD estimates from PPP processing depends strongly on the quality of satellite orbits and clocks. In our study, we focused on improving the double-difference processing because most EPN and E-GVAP analysis centres rely on a network approach utilizing double-difference observations, and many of them use Bernese GNSS Software v.5.2 (Dach et al., 2015).

The most widely used baseline strategy in double-difference processing is the so-called "obs-max" strategy which maximizes the number of double-difference observations from all stations of the network simultaneously (Dach et al., 2015). The optimization of the baselines is performed independently day after day. This strategy is especially convenient because it determines automatically the best baseline structure for every given day. The best stations (with most observations) are thus connected together to form the skeleton the network while the least stations are relegated to the peripheral, hence minimizing their detrimental effects (biases and gaps) on the other stations. The obs-max processing strategy leads theoretically to the most accurate estimates (ZTD, coordinates…) since it uses the maximum possible number of observations. This strategy has a drawback, however, which is to use slightly different network geometry every day as the number of observations changes. Day to day changes in the baseline geometry have an impact on the stability and repeatability of the estimates. To circumvent this effect, other baseline designs have been introduced. Jaworski et al. (2011) and Bosy et al. (2012) used a pre-defined network which contains a baseline skeleton of EPN reference stations which serves as the nodes to which national stations are connected in a star-like geometry. They show that this strategy leads to accurate and stable coordinates for a national network. In this study we show that both these strategies have limitations and are prone to ZTD outliers and gaps. Investigation of various case studies helped us to identify the weaknesses of these strategies. We propose and test an alternative baseline strategy that overcomes the most severe limitations and yields more stable ZTD time series with less outliers and gaps. We also describe an efficient outlier detection method for the final screening of the reprocessed ZTD time series and assess the quality of final ZTD data by comparison with ERA-Interim reanalysis.

Section 2 introduces the details of initial, standard, processing strategy which is used to calculate the station coordinates in national GNSS networks. In section 3, results from standard strategy are discussed and some case studies are shown. Section 4 describes the new, improved, baseline strategy. In section 5 and 6, the new baseline strategy is compared to the initial and obs-max strategies. Section 7 briefly describes the screening procedure used to remove the remaining outliers in the ZTD estimates and the solutions from the different strategies are objectively evaluated by comparison with ERA-Interim reanalysis. Section 8 concludes and draws perspectives.

## 2 Initial processing strategy

In this study, GPS data from 136 stations were analysed for year 2014. They include 104 stations of the Polish national Ground Based Augmentation System (GBAS) network – ASG-EUPOS, and 32 EPN stations (remote and in Poland – Fig. 1). The remote EPN stations (i.e. with baselines longer than 500 km) were included in order to provide absolute ZTD estimates (Duan et al., 1996; Tregoning et al., 1998). Details of the baseline strategy are given below. The processing was carried out for double-difference observations using Bernese GNSS Software v.5.2 (Dach et al., 2015). A minimum constraints approach was followed consistent with the general EUREF recommendations. The collected GNSS data were processed in 24-hour sessions starting at 00:00 UTC each day with data sampling of 60 second. The ionosphere-free linear combination of carrier-phase observations was used. An elevation cut-off angle of 3 degrees and elevation-dependent weighting of $cos^2z$ (with zenith angle $z$) were applied. In order to obtain the highest precision and accuracy, the IGS final precise satellite orbits, clocks, and earth rotation parameters were applied. Also, absolute antenna phase centre variations and offsets were used for ground and satellite antennas. All other systematic errors were modelled according to IERS current standards (Petit and Luzum, 2010). In addition, CODE's (Centre of Orbit Determination in Europe) global ionosphere maps were applied to minimize the impact of ionospheric delays on stations coordinates, including reduction of the second order ionospheric effects (Kedar et al., 2003). Also, the monthly differential code biases (DCBs) for satellites and receivers (P1-C1, P1-P2) were used. In the processing for all three strategies, a priori tropospheric delays were computed from the Global Pressure and Temperature (GPT) model combined with the Global Mapping Function (GMF) (Boehm et al., 2006) and Chen-Herring gradient model (Chen and Herring, 1997). The ZTD parameters were modelled as piecewise linear functions of time and estimated every 2 hours and the tropospheric gradients were estimated every 24 hours, with absolute and relative constraints of 5 m. The carrier phase ambiguities were resolved using the recommended procedure with Bernese software which applies several methods depending on the length of baselines, e.g. the SIGMA method and the Quasi-Ionosphere-Free (QIF) (Dach et al., 2015). For the all stations, daily observation files containing less than 1500 epochs (52% of expected epochs) were not used. The phase observations were screened with a threshold value of 4.0 mm (i.e. observations yielding post-fit residuals mapped to zenith above this threshold are eliminated for the final estimation).

The initial baseline design was based on common practice for the analysis of national GNSS networks. All the EPN stations were used as reference stations and connected together (Fig. 1a) and ASG-EUPOS stations were connected to the nearest EPN stations in a star strategy (Fig. 1b). With this design, all baselines are independent. It allows dividing the network in sub-networks which are processed independently with correct correlations and are combined afterwards (Dach et al., 2015). Starting with this initial baseline structure, the network is modified automatically by the Bernese software every day depending on the availability of observations. When observations for EPN station are missing for the whole day, the obs-max solution creates automatically new baselines with nearby stations (see e.g. Fig. 4 discussed later). This type of baseline design works well for positioning but has one major drawback for tropospheric monitoring. Indeed, when observations for a given station are missing only for a fraction of a day, the station will still be processed and all baselines connected to this station will be

impacted by the data gap. Since station coordinates are estimated daily, they are not much impacted by a short data gap (e.g. a few hours). On the other hand, since ZTD parameters are estimated every 2 hours, a data gap of about a few hours will strongly impact the ZTD estimates of all stations connected to the station with the gap. The situation becomes problematic when the gap produces a break in the network structure and a cluster of stations is disconnected from the main network (i.e. the one made of reference EPN stations including long baselines). This situation can arise in (i) star clusters composed of ASG-EUPOS stations which are connected to the main network via a single EPN-EPN baseline and in (ii) filaments and scattered clusters of stations created by obs-max as part of the automatic redesign of the baselines when no observations are available for some of the stations. Figure 2 shows an example of the first kind where station SULP is connected to the main network via USDL only. Other cases of this kind are: BYDG, ZYWI, and GANP. Figure 4 (discussed later) shows an example of the second kind when the initial EPN station BOGO has no observations and stations LOMZ and OSMZ are connected to MYSZ in a filament style. This type of situation is actually very common. The expected impact of disconnections of small clusters from the main network is a common mode bias in the ZTD estimates at the disconnected stations. Indeed, when the baselines are too short (<500 km), the ZTD estimates are highly correlated and absolute values cannot be properly recovered (Duan et al., 1996; Tregoning et al., 1998). The exact magnitude and sign of the bias is impossible to predict as it depends on the other errors in the observations, on the geometry of the baselines, and on the constraints applied on the ZTD parameters during the processing.

## 3 Results of initial processing strategy

The analysis of results from the initial processing strategy revealed many outliers due to disconnections of stations which showed up as spikes in ZTD and in formal error of ZTD. Here, we examine a few cases in order to quantify the magnitude of the biases and give an idea of their frequency.

Figure 2 illustrates the first kind of situation with the case of a group of five ASG-EUPOS stations connected in star configuration to EPN station SULP which itself is connected to the main network through a single baseline with EPN station USDL. Let us first focus on the spike at the end of day 226 (the errors at beginning of day 226 are of a different nature and will be discussed later). On day 226 the baseline USDL-SULP has observations only until 18:52 UTC. At that time station USDL has a data gap which lasts until day 230 at 05:46 UTC. From 18:52 UTC to the end of the day 226, the cluster composed of EPN station SULP and its five ASG-EUPOS stations is disconnected from the main network and common mode ZTD biases show up. The formal errors increase simultaneously as a result of high correlation between the estimated ZTD parameters. Table 1 reports the values for ZTD and formal errors for two of the ASG-EUPOS stations (BILG and CHEL) connected to SULP, as well as the values for SULP and USDL. The ZTD bias at BILG, CHEL, and SULP is varying as follows: +0.10 m at 20:00 UTC compared to 18:00 UTC, -0.14 m at 22:00 UTC compared to 20:00 UTC, and -0.26 m at 00:00 UTC (day 227) compared to 22:00 UTC. The next day, USDL is not available and the network is changed automatically. The cluster of stations connected to SULP is then connected to the main network through another baseline, and the ZTD estimates and formal errors

recover regular values. This kind of situation repeated 17 times throughout year 2014, during which station USDL had observation gaps lasting between a few hours and a few days. When the gaps extended over two or more days, the disconnections impacted only the first and last day of the period. In total, 25 days were impacted by these 17 observation gaps. Figure 3 shows the maximum ZTD variations and sigma values observed at these stations due to data gap periods at station

USDL in exactly the same configuration as illustrated in Figure 2 (note that only 19 days when SULP was available are shown). Common mode ZTD biases are observed in all the cases with amplitudes varying between ~1 cm and 30 cm. Significant formal errors variations are observed as well. Similar outliers were found in the estimates of stations connected to BYGD, ZYWI, and GANP (all belonging to the first kind of clusters). A solution to the problems of the first kind of clusters would be to ensure that each EPN station has several baselines to the main network.

The second kind of situation (initial reference network modified automatically by Bernese coincident with a gap in one of the connecting stations of a small cluster) is exemplified in Figure 4. In this case, reference EPN station BOGI is not available and obs-max strategy connected ASG-EUPOS stations OSMZ, LOMZ, and MYSZ in line. MYSZ is connected from the initial design to EPN station LAMA. On day 73 a data gap at station LAMA between 00:00 UTC and 11:00 UTC causes a disconnection of the chain of stations MYSZ-LOMZ-OMSZ. A common mode bias in the ZTD estimates of ± 10 cm is

observed at all three stations along with an increase in their formal errors. This problem repeats for MYSZ, LOMZ, and OMSZ for 17 days when LAMA has gaps and BOGI is not available, and also at other stations in similar configurations. A solution to the problems of this kind would be to impose that ASG-EUPOS stations are only connected to reference EPN stations and not between them.

Now let us come back to the spikes observed on beginning of day 226 (Fig. 2). Two sites (BILG and HRUB) show variations

in ZTD of 15-20 cm between 00:00 and 02:00 UTC and 5-10 cm between 02:00 and 04:00 UTC. One site has no ZTD estimates (SHAZ) and the others show smaller variations which are more in the expected range of values. During this period, no disconnection was observed from station USDL, so the origin of the biases must be different. Inspection of post-fit residuals showed that during this period, few double difference observations were available for the baselines connected to SULP (1-4 satellites were observed at a time for BILG-SULP baseline, 1-3 satellites for HRUB-SULP, and 3 satellites for HOZD-SULP).

Moreover, a lot of ambiguity parameters were estimated over this period which might be due to many short data gaps at SULP. The formal errors in the lower plot of Fig. 2b reflect the differences in the number of observations. The biases in ZTD estimates can only be explained by increased errors during processing, e.g. ambiguity parameters not properly fixed (Dousa, 2002) or errors in satellite orbits (Dousa, 2010). It is clear that the larger biases are connected with the smaller number of used satellites (i.e. fewer observations, but also weaker geometry) and increased number of ambiguities. This kind of spikes, not due to

disconnections from main network, but to data gaps at a station, was observed for many stations. Figure 5 shows the number of observations per day at SULP and two other EPN stations, ZYWI and KATO, which are good stations (their mean numbers are around 22,000 while the best stations reach mean numbers around 23,000). Station SULP has a mean of 16,547. This was the smallest number among all Polish EPN stations. It was thus decided to remove SULP from the new solution (section 4). A

threshold on the number of observations per day of 14,000 was also introduced for the reference EPN stations to limit the impact of their data gaps on the connected ASG-EUPOS stations. This test was applied daily.

In addition to the problems described above, we also noticed that in case of gaps in observations, spikes in ZTD and formal error often appear at the edges of the gaps. For example, station USDL has a data gap between day 226 at 18:52 UTC and day 230 at 05:46 UTC. The ZTD estimates at the edges of the gap (day 226 at 20:00 UTC and day 230 at 04:00 UTC) are expected to be less accurate because of fewer observations. Table 1 shows that last ZTD estimate on day 226 is at 20:00 UTC which value differs from the previous one by more than 4 cm while its formal error increases from 1.2 mm to 3.2 mm. The next ZTD estimate is for day 230 at 4:00 UTC which value differs from the next one by 3 cm and its formal error is 12.4 mm. Though ZTD variations of 3 cm or 4 cm in 2 hours are possible, they are very unlikely in this case based on the observed variations for the preceding and following ZTD values. We thus decided to remove systematically the ZTD estimates at the edges of gaps in the post-processing stage to avoid this kind of spikes.

## 4 New baseline strategy

A new baseline strategy was developed to fix the problems which appeared in the initial solution and to ensure that all the stations remain connected to the main reference network. Therefore, the main reference network composed of local EPN stations from Poland and nearby countries was first optimized every day using Bernese obs-max strategy. This ensured that the EPN stations with small numbers of observations are relegated to the peripheral of the network and thus limit the risk of disconnections as observed in the initial strategy. Then the remote EPN stations were connected to this local reference network to strengthen the decorrelation between ZTD parameters and provide absolute ZTD estimates. For this purpose, 15 high quality remote EPN stations were chosen based on statistics and information available on the EUREF server. We decided to discard two EPN stations (SULP, UZHL), because of to their small number of observations in 2014. Also, we chose station BOGO instead of BOGI as a reference EPN station in Poland due to better quality and stability of BOGO. Finally, we set a threshold of 14,000 for the number of double difference observations on EPN stations and removed stations below this limit from the daily solution to minimize the risk of disconnections. This number was chosen after numerous tests and appeared as a good compromise between number of removals and baseline lengths. The final procedure carried out for each daily session separately is the given below and illustrated with an example in Figure 6:

1. Selecting the reference EPN stations in Poland and near countries, creating the reference network based on the results of a preliminary processing with Bernese using obs-max strategy, and removing those stations which have fewer than 14,000 observations.

2. Connecting remote EPN stations to the reference network at peripheral stations (bad stations) which have only one baseline (e.g. BOR1, LAMA, USDL) using a shortest baseline approach, to reduce the risk of disconnections of parts of the stations.

3. Connecting additional remote EPN stations to Polish EPN stations with the highest number of observations (the best stations) to strengthen the decorrelation of ZTD estimates (e.g. BYGD, JOZE, SASS).

4. Finally, connecting the ASG-EUPOS stations to the reference EPN network using the shortest baseline approach and a „star" structure.

## 5 Comparison of results from initial and new baseline strategies

Here, we compare the results from initial (old) and new baseline strategies. The ZTD estimates at the edges of gaps were removed beforehand. Figure 7 shows the results for station BILG. Overall, it is seen that most spikes in ZTD and formal error present in the old solution are avoided with the new strategy (Fig. 7a). Now let us focus on day 226 (Fig. 7b). With the new strategy, BILG is connected to EPN station USDL. Station USDL has good observations, with no interruptions in measurements at the beginning of the day contrary to station SULP to which BILG was connected in the old solution (Fig. 2). The spikes at beginning of the day seen at BILG in the old solution are thus avoided. However, USDL has a data gap from 18:52 UTC until the end of the day. After 20:00 UTC, station BILG has thus no more ZTD estimates. The point at 20:00 UTC is also removed since it precedes a gap. The large spikes seen in the old solution after 18:00 UTC because of disconnection of SULP and BILG are also avoided. On day 227, when USDL is not available, BILG is connected to KRA1 and the old and new solutions are fairly consistent.

Another example is shown in Figure 8 with station KUZA. In both solutions, this ASG-EUPOS station is connected to EPN station ZYWI. Overall, all spikes in ZTD except the one on day 102 are removed in the new solution (Fig. 8a). On day 102, the spike in ZTD is due to a data gap at ZYWI between 00:12 and 03:50 UTC. Since ZYWI had 17,742 observations on that day it was not removed and both the old and new solution estimated exactly the same ZTD values. Figure 8b shows a period when ZTD spikes are effectively removed. During days 26 to 30, ZYWI had many data gaps with maximum numbers of observations of $11,190 - 10,686 - 0 - 6,615$ and $12,407$ respectively, on these five days with the old strategy. As a consequence, large formal errors are observed at KUZA on days when the station is connected to ZYWI (all except day 28). The two large spikes in ZTD seen at end of day 27 and beginning of day 30 are again due to very few observations in common with ZYWI (for 10-15 minutes only) on 1 or 2 satellites only at a time (similar to day 102). However, with the new strategy, the baseline KUZA-ZYWI is not used on these days because of too few observations (below 14,000). Instead, KUZA is automatically connected with EPN station KATO and the resulting ZTD and formal error time series are much smoother.

With the new baseline strategy, ASG-EUPOS stations are only connected to reference EPN stations and all these EPN stations have at least two baselines with the main reference network (local and remote EPN stations). Disconnection of clusters (Fig. 2) or chains of stations like MYSZ-LOMZ-OSMZ (Fig. 4) are avoided by construction. The only spikes remaining in the ZTD series in the new solution are thus due to small number of observations. We tested the idea of using constraints between successive ZTD parameters in order to smooth the ZTD time series and thus reduce the outliers. The idea works in general as both resulting ZTD variations and formal errors stay in a range consistent with the imposed constraints (e.g. 0.1 m), but outliers

can still be detected in the time series. Using tighter constraints would further smooth outliers but also the variations in ZTD due to atmospheric variability. This is thus not a good solution to remove the remaining outliers and only the use of a proper screening method can help (see section 7).

## 6 Comparison to obs-max solution

Let us now quantify in a more statistical way the improvement of the new strategy compared to the old one and compare them both to the standard obs-max strategy taken as a reference. To this purpose, the same network has been reprocessed with Bernese software using the obs-max strategy for all stations (not distinguishing between EPN and ASG-EUPOS, or between Polish and remote stations). As a measure of the quality of each of three solutions, we computed the standard deviations of the ZTD forward differences and of the formal errors for each station. Taking the forward differences of 2-hourly ZTD values

allows to remove almost completely the seasonal variations and at the same time to magnify the outliers. Large standard deviations are thus symptomatic of time series containing outliers. However, in order to limit the impact of too large outliers (sometimes as extreme as -1 m and +5 m) we first applied a "light screening" composed of a range check on ZTD, with lower and upper limits of 0.5 m and 3.0 m, and on formal error, with an upper limit of 0.1 m. The number of removed values is given in Table 2. It represents less than 0.03% of all ZTD values.

Figure 9 shows the distribution of standard deviations of ZTD forward differences and of formal errors for 104 common ASG-EUPOS stations processed in all three solutions. Table 2 reports the mean values over all stations. We can see from Figure 9 and Table 2 that the ZTD variations and formal errors are smaller, i.e. solutions are more stable and more accurate, for the new and obs-max solutions. Maybe surprisingly, the mean ZTD variability is smaller for the new solution compared to obs-max. However, the mean formal error is the smallest for obs-max (this is consistent with the fact that this strategy maximizes the

number of observations). Obs-max provides also the largest number of ZTD estimates. However, the new solution achieves smaller ZTD variability and formal errors than obs-max at most sites (Table 2 reports the numbers of sites for which each solution is the largest among all: e.g., the new solution has the largest ZTD variations at only 11 sites, whereas obs-max has the largest number at 31 sites and the old solution at 62 sites). An explanation is given below.

   Large variability in ZTD and formal error (Figure 9) are observed with all three solutions at 3 sites: KOSZ, WLAD, and SHAZ.

Inspection of time series shows many spikes in ZTD for these sites which are in general associated with low number of observations. Obs-max has also bad results for station OPLE, but these are due to large ZTD spikes on one specific day (17 January, not rejected by the "light screening") when the station has many small data gaps. In general, the spikes in ZTD are coincident with spikes in formal error which can be detected and removed during the final screening step (section 7).

   The fact that the new solution provides better results than obs-max was investigated in more detail for special cases when

outliers appeared in the obs-max solution that were not present in the new solution. One example is for station GDAN (Fig. 10). Spikes in ZTD and formal error are observed in obs-max solution on days 170-171, but not in the new solution. The number of epochs and observations collected at GDAN was high, so it is not the same case as described above for KUZA,

KOSZ, WLAD or SHAZ. Inspection of the design of the network in both processing variants (Fig. 11) shows that in obs-max solution, station GDAN was automatically connected to station WLAD and WLAD to EPN station REDZ, while in the new solution station GDAN was connected to EPN station REDZ (star structure). Table 3 shows the number of processed observations for the mentioned baselines with the two processing variants on day 170, before and after residual screening carried out automatically by Bernese software during processing. The numbers of observations are large before the screening for all the baselines, so there was no significant observation gap on that day. However, the numbers dropped strongly after screening which reveals that observations were of bad quality, especially for station WLAD, and to lesser extent for station GDAN. In this situation, the baseline design chosen by obs-max based on a priori number of observations revealed not the best a posteriori. We counted 83 days of this kind in 2014 for station GDAN. A solution to this problem would be to optimize the baselines based on post-residual screening statistics. In the new solution, a preliminary selection of reference stations was applied and ASG-EUPOS stations were connected only to EPN stations which had more than 14,000 of used observations.

## 7 ZTD screening and comparison to ERA-Interim

Despite the new processing strategy allows to produce more stable and more accurate ZTD time series in comparison to the initial and obs-max strategies, a few outliers may still remain due to short data gaps or increased errors at the stations of a baseline, even if the first and the last ZTD estimates around observation gaps are systematically removed (as discussed earlier). The goal of the screening procedure described below is to detect and remove these outliers. Following the approach proposed by Bock et al., (2014), the procedure consists in applying first a range check on the ZTD and formal errors $\sigma_{ZTD}$ to remove the values that are physically out of range. As a second step, an outlier check is applied where the thresholds are computed from the data themselves for each station (this is a main reason why the out-of-range values must be removed beforehand). Several variants of range check thresholds and outlier check limits were tested. In addition to the "light screening" mentioned in section 6, three other variants are presented here.

Screening variant No. 1

- range check on ZTD: remove values outside of the interval [2.0 m ; 2.6 m]
- range check on formal error: remove values with $\sigma_{ZTD} > 1$ cm

Screening variant No. 2

- range check on ZTD and formal error as in variant No. 1
- sigma outlier check: remove values with $\sigma_{ZTD} > \text{median}(\sigma_{ZTD}) + 3.5 \times \text{std.dev.}(\sigma_{ZTD})$

Screening variant No. 3

- range check and outlier check as in variant No. 2
- 00:00 UTC values removed to avoid day boundary effects.

The results from these three screening variants are shown in Table 4. It is seen that in all cases, the new solution achieves smaller ZTD variability and formal errors than obs-max at most sites (column 2 and 3), the number of rejected ZTDs (and thus

the number of used ZTDs) are very similar between the new and obs-max strategies (columns 4 and 5), and the mean standard deviations of ZTD and formal errors (columns 6 and 7) are slightly smaller for the new solution with screening variants No. 1 and No. 2. The stability of the ZTD time series (column 6) is improved (by ~5%) with variant No. 1 compared to the "light screening". The improvement is more spectacular in terms of formal error (60-70%). This is a good indicator of the efficiency

of outlier rejection. Screening No. 1 and No. 2 reduced significantly the standard deviation of ZTD forward differences and formal errors for stations KOSZ, SHAZ, and WLAD (Fig. 12) which were pointed previously as bad stations. Overall, screening variant No. 2 removes about 1.2% of the ZTD estimates which remains at an acceptable level when high accuracy is searched.

Screening No. 3 was introduced to assess the weight of the day boundary effects in the overall statistics. When all 00:00 UTC

estimates are removed, the stability and accuracy of ZTD estimates is significantly improved (Table 4). This screening option is however not to be used as it removes useful data. A better solution to the day boundary problem would be to combine solutions from successive days at the normal equation level (Dousa et al., 2017). The combination adjusts ZTD estimates across the 00:00 UTC boundaries for the central day and minimizes discontinuities between days.

As a final validation step, GPS ZTD estimates were compared to ERA-Interim reanalysis (Dee, et al., 2011). The reanalysis

ZTD data were adjusted for the height difference between the model topography and the GPS stations. The GPS ZTD data were screened with screening variant No. 2. Mean and standard deviation of ZTD differences between GPS and ERA-Interim and correlation coefficients of 6-hourly time series are shown in Figure 13. The results are shown for 103 common ASG-EUPOS stations (station WAT1 is not displayed here because of a bias of -2 cm of unknown origin). All three processing variants are fairly consistent compared to ERA-Interim ZTD estimates. This suggests that the differences are either due to

common-mode GPS errors, representativeness differences, or errors or in the reanalysis. GPS errors are suspected at stations with large ZTD differences (HOZD, MIEL, SHAZ, SKSV, WLAD). Some of them were already detected as problematic in the previous sections from the inspection of ZTD variability and formal errors (Fig. 9) or from the inspection of screening results (Fig. 12).

Table 5 reports the mean values over all stations for the three variants. The mean differences are around -3.0 mm pointing to

25 a slight moist bias in ERA-Interim or a slight dry bias in GPS. At this level of accuracy it is difficult to say which of ERA-Interim or GPS is biased in an absolute sense. We note however, that the sign and magnitude of the bias are consistent with other studies (Dousa et al., 2017). The standard deviations of differences are about 10-11 mm, which is significantly larger than differences between GPS solutions (as also noticed in previous studies, e.g. Dousa et al., 2017). The mean correlation coefficients are about 0.975 for all three variants. These high correlations are mainly dominated by the large seasonal

variations. It is not straightforward to distinguish which is the best strategy based on the mean values reported in Table 5. However, from Figure 13 it is clear that the new strategy is often better than the two others based on lower standard deviation of differences and higher correlation. Counts given in Table 5 show that the new strategy leads to larger standard deviation of differences for only 3 stations when all three variants are compared or 9 stations when only the new and obs-max are compared.

Regarding correlations, the new strategy leads to lower correlations in only one case when all three variants are compared or six cases when only the new and obs-max are compared.

## Conclusions

This study aims at understanding the main factors leading to outliers in GPS ZTD time series in a sub-regional network (typically a permanent national GNSS network). We show that the baseline design strategy in a double-difference network processing has a strong impact on the quality and continuity of ZTD time series. ZTD outliers are most of the time caused by sub-daily data gaps at reference stations which provoke disconnections of clusters of stations from the reference network and common–mode biases due to the strong correlation between stations in short baselines. We developed an alternative baseline strategy that minimizes such disconnections and yields more stable ZTD time series with less outliers and gaps. The new strategy ensures that all the stations remain connected to the main reference network. The reference network is optimized for each daily session separately using Bernese obs-max strategy and reference stations are removed from processing if their daily number of observations is lower than 14,000 (this is an empirical limit which can be adjusted). With the new baseline strategy, the stations of the sub-regional network (in our case ASG-EUPOS in Poland) are only connected to reference EPN stations and all these EPN stations have at least two baselines with the main reference network (to local and remote EPN stations), consequently disconnections of clusters or chains of sub-regional stations are avoided by geometry of the network. The only spikes remaining in the ZTD series in the new solution are due to small number of observations or short gaps at sub-regional stations. They are removed in a post-processing screening procedure which consists in: 1) the removal of the first and the last ZTD estimates around observation gaps, 2) range check and outlier check on ZTD and formal errors. The range-check and outlier check detect spikes in ZTD and formal errors based on constant and station-specific thresholds, respectively. The screening removed about 1.2% of ZTD estimates which remains at an acceptable level when high data continuity is searched. Finally, screened GPS ZTD estimates were compared to ERA-Interim reanalysis to assess the quality of final ZTD data and detect smaller bias and jumps.

We investigate the cases when the new strategy provides better results than obs-max solution. Although obs-max maximizes the number of double-difference observations from all stations of the network simultaneously, the baseline design is chosen by obs-max based on a priori number of observations. In case of bad quality of observations, the number may drop strongly after residual screening carried out during processing. This leads to more ZTD outliers. A solution to this problem is to optimize the baseline based on post-residual screening statistics and apply a preliminary selection of the reference stations. This is done in the new baseline strategy.

Precise Point Positioning (PPP) might be an interesting alternative to outliers arising from defects in the baseline geometry in a double-difference processing. PPP is based on single station observations, meaning that no baselines between stations are computed. Then, there is no problem of common mode biases when there are observation gaps in nearby stations. However, PPP is mainly affected by the quality of orbits and clocks for which very accurate products are not available in real time (e.g.

for now-casting weather application). This is one of reasons why most of E-GVAP analysis centres use double-difference processing while the dependency on the clock products is much smaller.

The improved processing strategy may be also an interesting approach for reprocessing historical data to generate a new data with less outliers or to the operational processing to improve future ZTD estimates. More accurate and stable ZTD series may be produced in this mode, and the impact of equipment changes may be more easily detected in the double difference residuals than in zero difference residuals. Some scientific applications also use GNSS tropospheric gradient estimates (Dousa et al;, 2017) which were not considered in this study. We argue that similar kind of outliers probably affect the gradient estimates and that the processing strategy proposed in this work would also similarly reduced them. The analysis of gradients and long-time series will be considered in a future work.

## Acknowledgements

This work has been supported by Polish National Science Centre grant No. UMO-2015/19/B/ST10/02758. The study was partially carried out during Short Term Scientific Mission (STSM) in the framework of ES1206 COST Action.

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

**Table 1: Estimated ZTD and formal error (sigma) at stations impacted by a gap in observations at reference station USDL at 20:00 on day 226 (see also Figure 2).**

| STATION | day HH:MM | ZTD [m] | sigma [m] |
|---------|-----------|---------|-----------|
| **BILG** | 226 18:00 | 2.4249 | 0.0023 |
| | **226 20:00** | 2.5288 | 0.0050 |
| | 226 22:00 | 2.3827 | 0.0104 |
| | 227 00:00 | 2.1211 | 0.0296 |
| | 227 00:00 | 2.4108 | 0.0031 |
| | 227 02:00 | 2.3894 | 0.0018 |
| **CHEL** | 226 18:00 | 2.4314 | 0.0021 |
| | **226 20:00** | 2.5196 | 0.0048 |
| | 226 22:00 | 2.3965 | 0.0100 |
| | 227 00:00 | 2.1132 | 0.0291 |
| | 227 00:00 | 2.4063 | 0.0029 |
| | 227 02:00 | 2.3950 | 0.0018 |
| **SULP** | 226 18:00 | 2.4205 | 0.0019 |
| | **226 20:00** | 2.5273 | 0.0049 |
| | 226 22:00 | 2.3946 | 0.0107 |
| | 227 00:00 | 2.0855 | 0.0321 |
| | 227 00:00 | 2.3698 | 0.0029 |
| | 227 02:00 | 2.3610 | 0.0017 |
| **USDL** | 226 14:00 | 2.4087 | 0.0010 |
| | 226 16:00 | 2.3902 | 0.0010 |
| | 226 18:00 | 2.3938 | 0.0012 |
| | **226 20:00** | 2.3561 | 0.0032 |
| | **no obs** | - | - |
| | 230 04:00 | 2.3296 | 0.0124 |
| | 230 06:00 | 2.2942 | 0.0008 |
| | 230 08:00 | 2.2891 | 0.0008 |

**Table 2: Statistics of ZTD estimates and formal errors (sigma) for all three processing variants discussed in the text, computed over 104 common ASG-EUPOS stations. The best values are indicated in bold. Column 2 (resp. 3) gives the number of stations for which the standard deviation of ZTD (resp. sigma) is maximal among the three solutions (e.g. standard deviation of ZTD of the old solution is maximal 62 times out of 104).**

| Solution | Times Max STD(ZTD) | Times Max STD(sigma) | Rejected data | Used data | Mean STD(ZTD) | Mean STD(sigma) |
|---|---|---|---|---|---|---|
| "light screening": range check on ZTD [0.5 m; 3.0m], on sigma [0 m; 0.1m] | | | | | | |
| old | 62 | 81 | 148 | 468332 | 0.0142 m | 0.00119 m |
| **new** | **11** | **6** | **84** | 469534 | **0.0129 m** | 0.00079 m |
| obs-max | 31 | 17 | 109 | **471666** | 0.0133 m | **0.00067 m** |

**Table 3: Number of processed observations with the two processing variants (obs-max and new), on day 170, before and after residual screening carried out automatically by Bernese software.**

| Obs-max strategy | | New strategy | |
|---|---|---|---|
| Baselines | Nb of observations | Baselines | Nb of observations |
| Before residual screening | | | |
| GDAN-WLAD | 29,806 | GDAN-REDZ | 30,571 |
| WLAD-REDZ | 30,933 | WLAD-REDZ | 30,933 |
| After residual screening | | | |
| GDAN-WLAD | 6,942 | GDAN-REDZ | 17,072 |
| WLAD-REDZ | 6,774 | WLAD-REDZ | 6,774 |

**Table 4: Similar to Table 2 but for other screening variants. The numbers in brackets in columns 6 and 7 indicate relative difference with the results of the "light screening" (values given in Table 2).**

| Solution | Times Max STD(ZTD) | Times Max STD(sigma) | Rejected data | Used data | Mean STD(ZTD) | Mean STD(sigma) |
|---|---|---|---|---|---|---|
| **screening 1: range check on ZTD [2.0m, 2.6m], on sigma [0 m; 0.01m]** | | | | | | |
| old | 54 | 84 | 1453 | 466487 | 0.0129 m (-9%) | 0.00040 m (-67%) |
| **new** | **7** | **2** | **668** | 468705 | **0.0124 m (-4%)** | **0.00031 m (-61%)** |
| obs-max | 43 | 18 | 696 | **470824** | 0.0127 m (-5%) | 0.00031 m (-53%) |
| **screening 2: screening 1 + outlier check on sigma > (median(sigma) + 3.5*STD(sigma))** | | | | | | |
| old | 45 | 90 | 6700 | 459637 | 0.0126 m (-11%) | 0.00025 m (-79%) |
| **new** | **8** | **0** | 5279 | 462658 | **0.0123 m (-5%)** | **0.00021 m (-73%)** |
| obs-max | 50 | 14 | **5198** | **465187** | 0.0125 m (-6%) | 0.00023 m (-65%) |
| **screening 3: screening 2 + 00 UTC values removed** | | | | | | |
| old | 40 | 93 | 76718 | 354188 | 0.0109 m (-23%) | 0.00015 m (-88%) |
| **new** | **7** | **1** | **75708** | **357770** | 0.0110 m (-15%) | 0.00013 m (-84%) |
| obs-max | 57 | 10 | 76115 | 356274 | **0.0106 m (-20%)** | **0.00012 m (-82%)** |

**Table 5: Mean statistics of ZTD differences and correlations between GPS and ERA-Interim computed over 103 common ASG-EUPOS stations. Column 5 (resp. 6) gives the number of stations for which the standard deviation of ZTD difference (resp. correlation) is maximal (resp. minimal) among the three solutions. The numbers in brackets are for the comparison of new and obs-max only. The best values are indicated in bold.**

| Solution | Mean diff. ZTD (m) | Std. diff. ZTD (m) | Correlation | Times Max STD($\Delta$ZTD) | Times Min correlation |
|----------|--------------------|--------------------|-------------|----------------------------|-----------------------|
| old | **-0.0028** | 0.0106 | 0.9753 | 36 | 48 |
| **new** | -0.0030 | **0.0104** | **0.9766** | **3 (6)** | **1 (6)** |
| obs-max | -0.0032 | 0.0107 | 0.9749 | 64 (94) | 54 (97) |

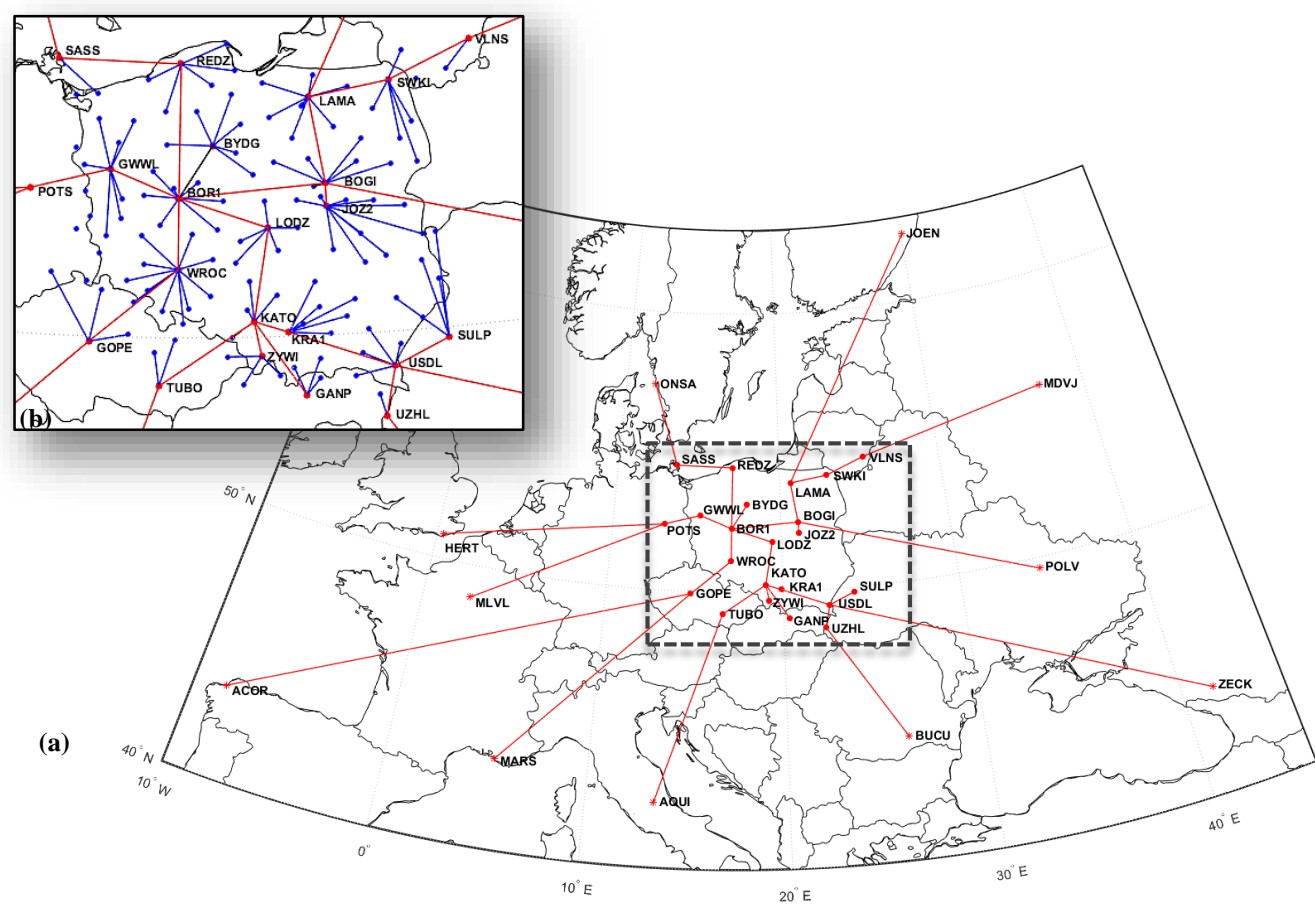

**Figure 1 (a) Map of reference EPN stations used for processing, comprising 22 EPN stations in Poland and near countries and 11 remote stations. (b) Zoom on ASG-EUPOS stations (blue dots) and EPN reference stations (red dots). The ASG-EUPOS stations are connected to the nearest EPN station in star network in Poland (blue lines).**

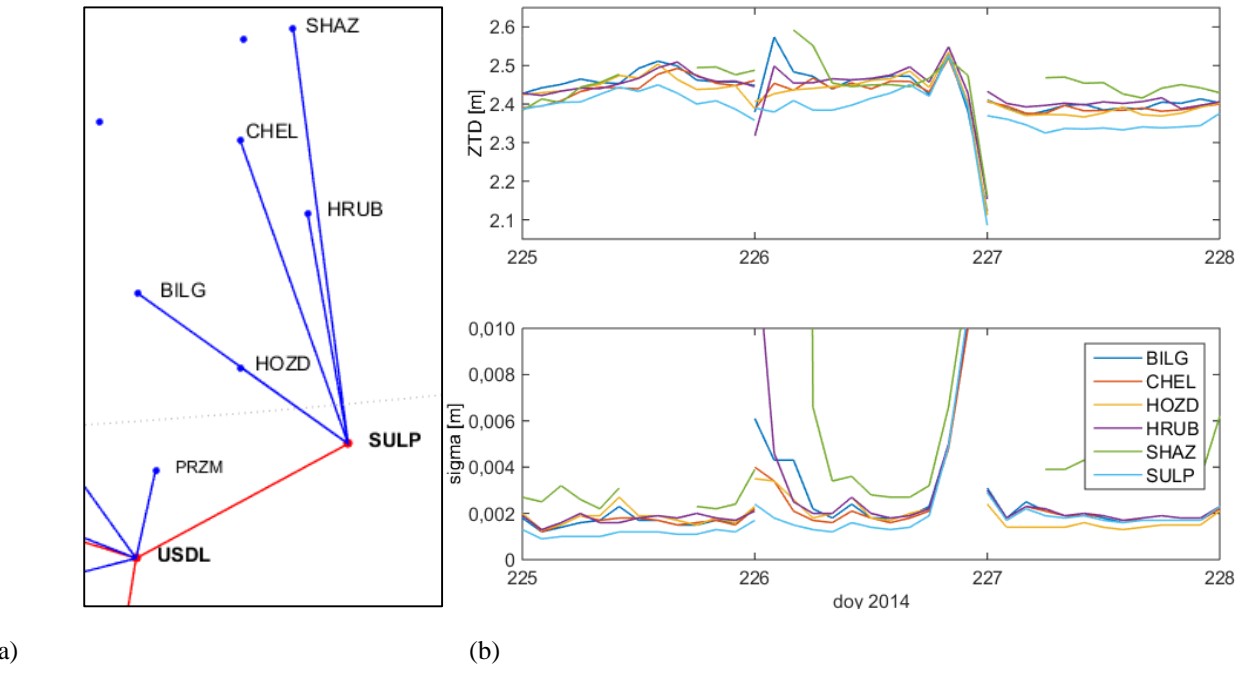

(a)                                                    (b)

**Figure 2: Example of common mode biases in ZTD affecting a cluster of stations (SULP, BILG, CHEL, HOZD, HRUB, SHAZ) due to disconnection from the reference network. (a) In this network design the cluster is connected to the main network through a single baseline SULP-USDL. (b) A data gap at station USDL on day 226 after 18:52 UTC provokes a common mode bias at all stations of the cluster.**

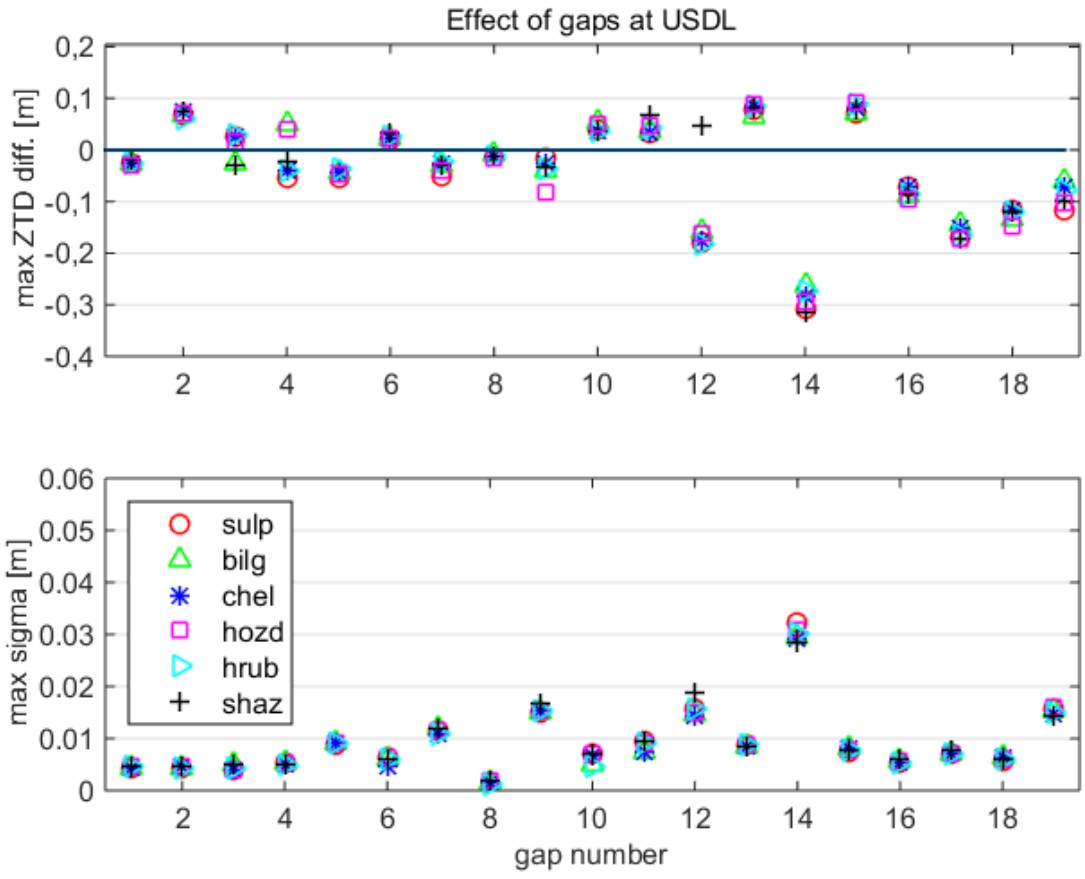

**Figure 3: Maximum ZTD variation (positive or negative) at stations of the cluster based on SULP during data gap periods at station USDL (gap number 13 corresponds to day 226 on Figure 2). Below, corresponding formal error.**

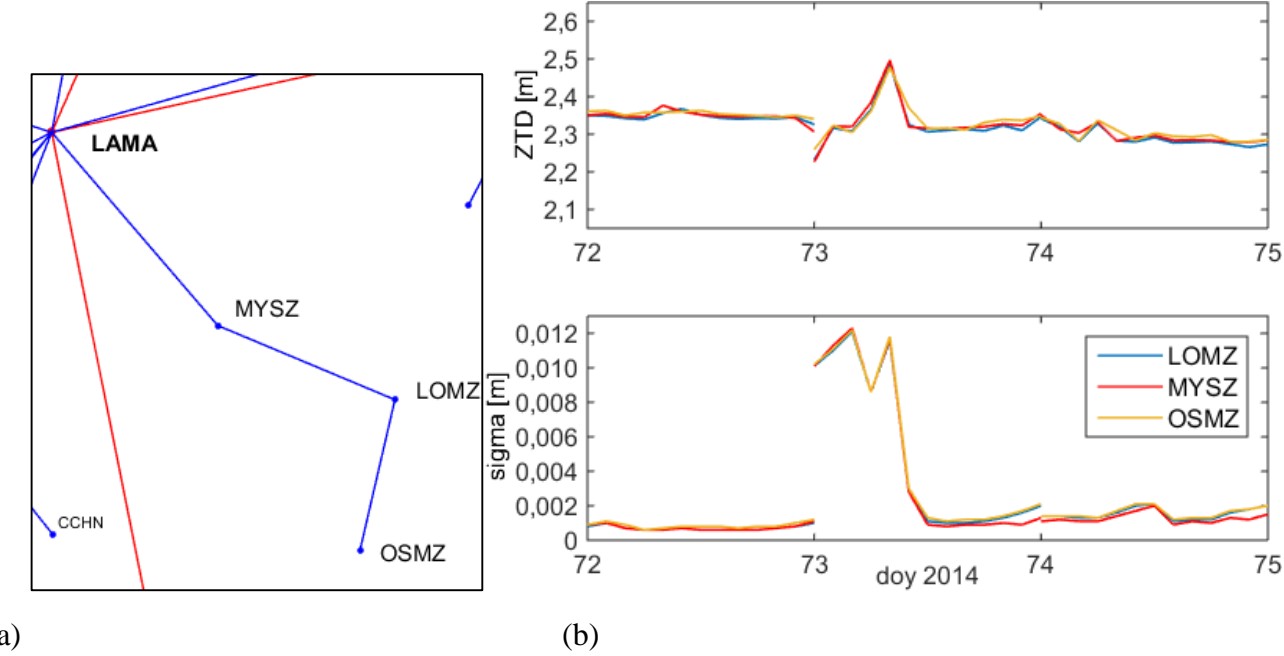

(a)                                                         (b)

**Figure 4: Example of common mode biases in ZTD affecting a cluster of stations (MYSZ, LOMZ, OSMZ) due to disconnection from the reference network. (a) In this network design the group of stations is connected to the main network through a single baseline MYSZ-LAMA. (b) A data gap at station LAMA on days 73 provokes a common mode bias at all stations of the group.**

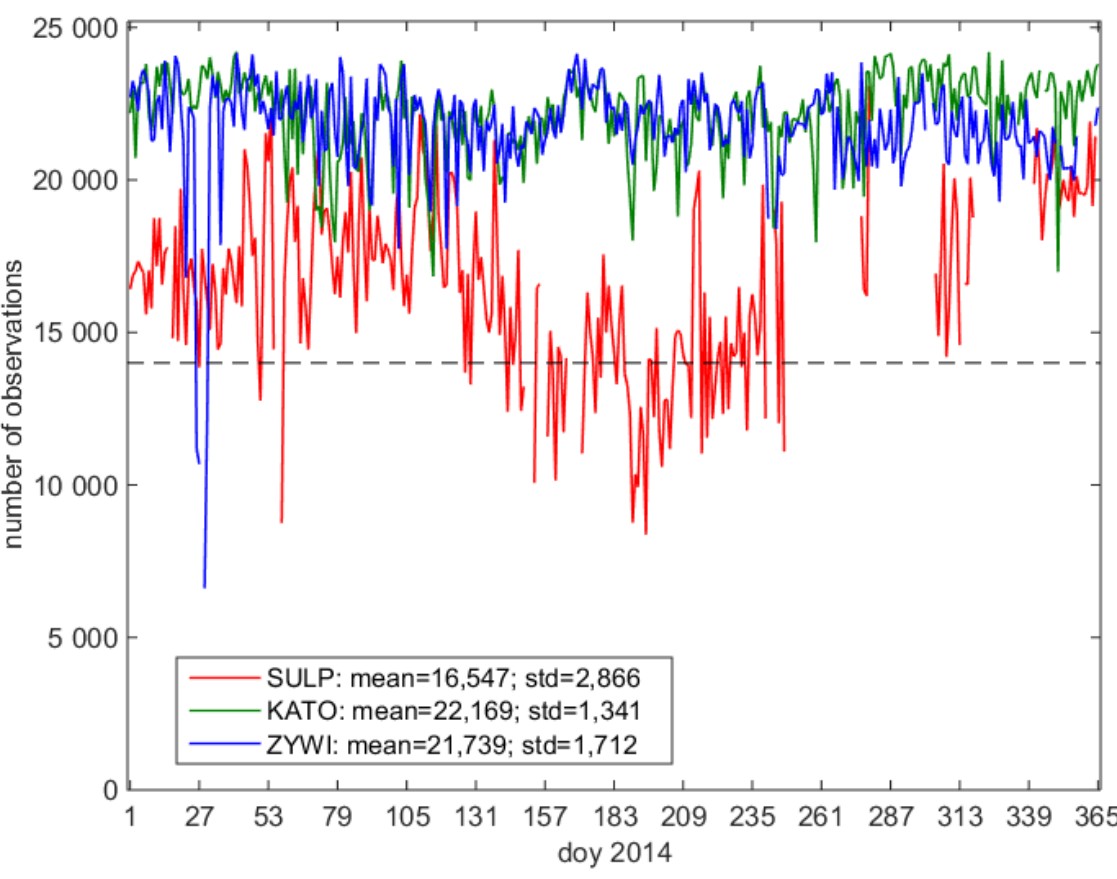

**Figure 5 : Time series of the number of observations (or maximum when the station is involved in several baselines) for stations SULP, KATO, and ZYWI. The dashed line shows the threshold used to eliminate days with low numbers of observations.**

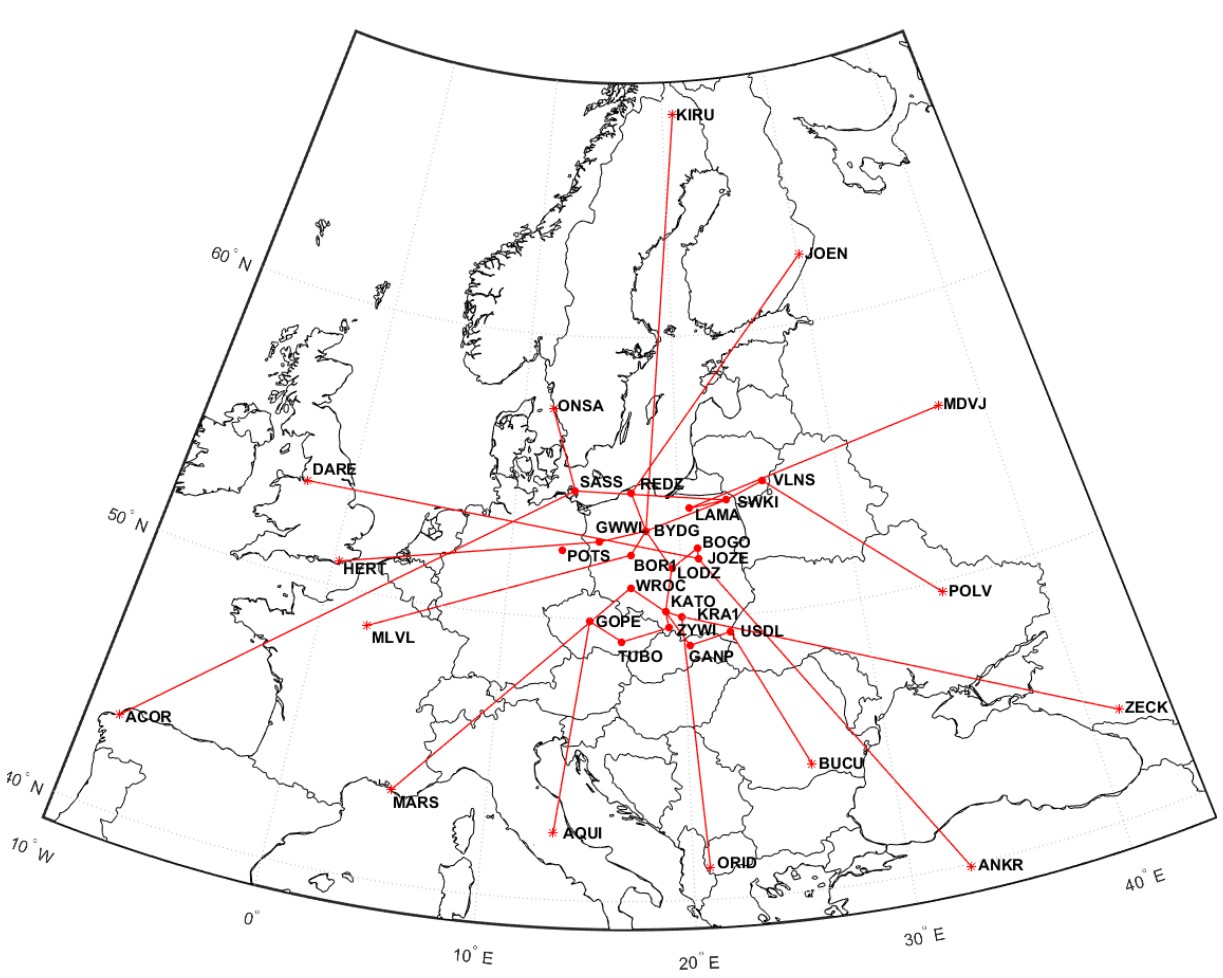

**Figure 6: Map of reference network composed of EPN stations from Poland and nearby countries created using Bernese obs-max strategy on day 1 of year 2014, with the new baseline strategy. Remote stations are connected to the local reference network to strengthen the decorrelation between ZTD parameters.**

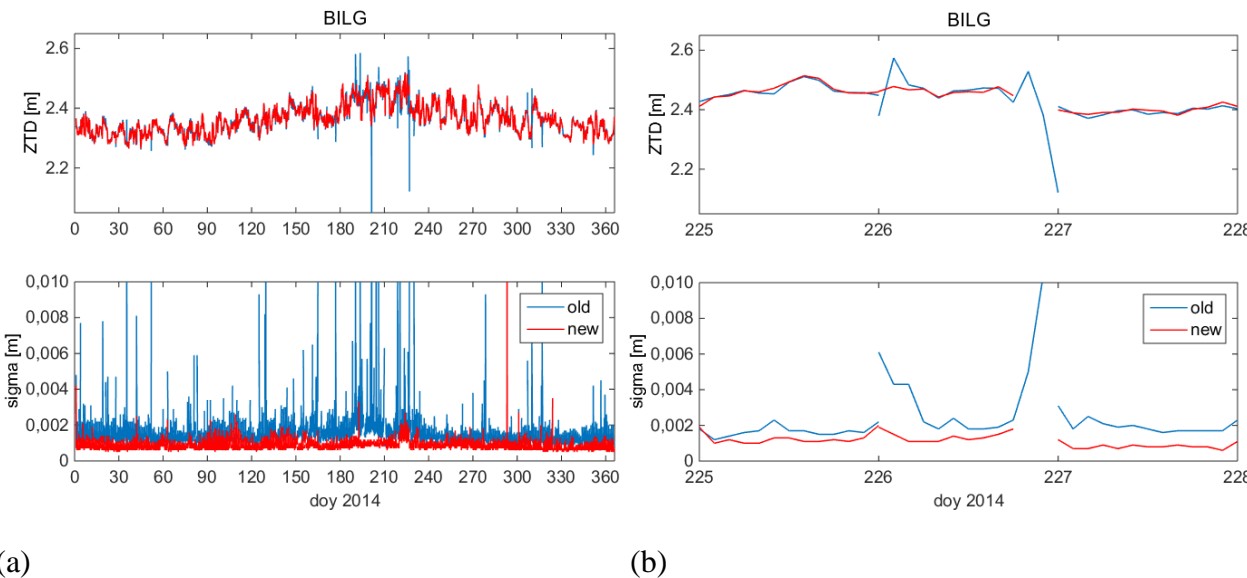

(a)                                                   (b)

**Figure 7: Comparison of ZTD estimates and formal error for the old (blue lines) and new (red lines) baseline strategies. a) Full time series for station BILG. b) Zoom on period when the old solution has outliers due to a reduction in the number of observations (beginning of day 266) and a gap in the observations (end of day 226) at reference station USDL. The new solution has only a short gap but no outliers.**

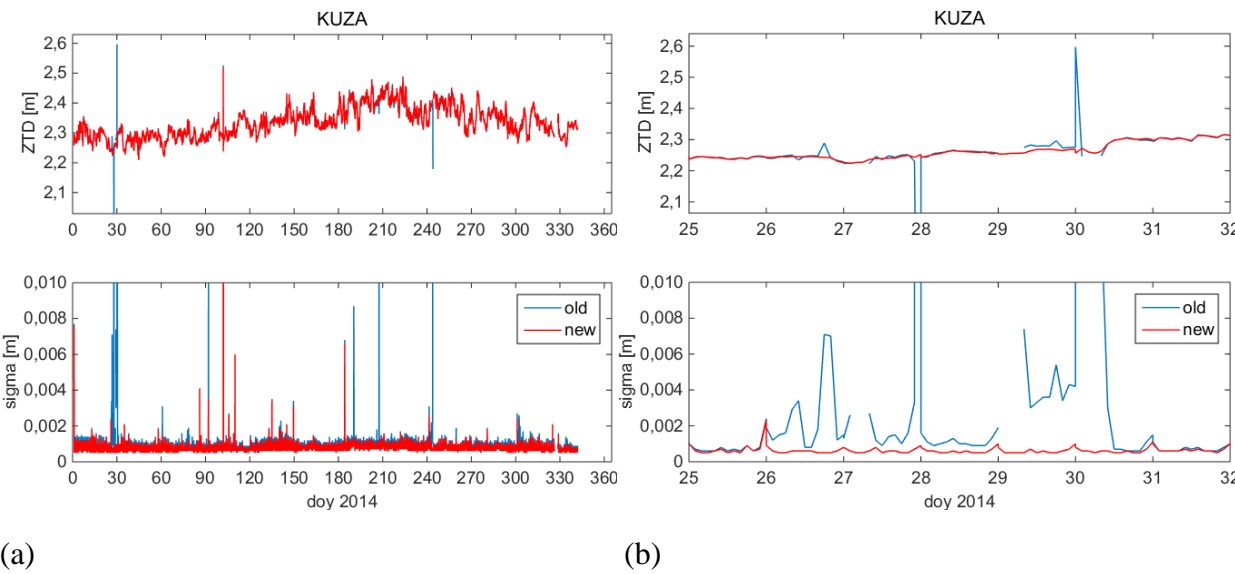

(a)                                                                     (b)

**Figure 8: Similar to Figure 7 for station KUZA. a) Full time series. b) Zoom on a period when the reference station ZYWI has many observation gaps which cause outliers in old solution. In the new solution station KUZA is not impacted.**

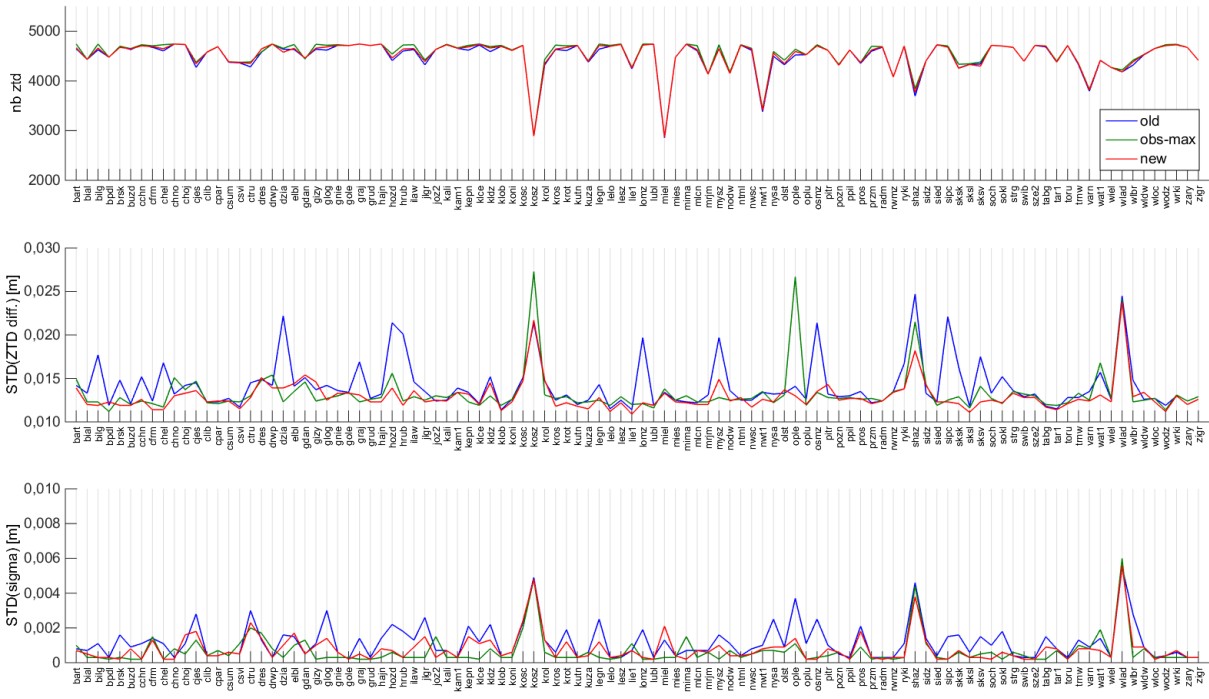

**Figure 9: Distribution of standard deviations of ZTD differences and formal error of ZTD for the 104 common ASG-EUPOS stations: old solution – blue line, obs-max– green line, new solution – red line.**

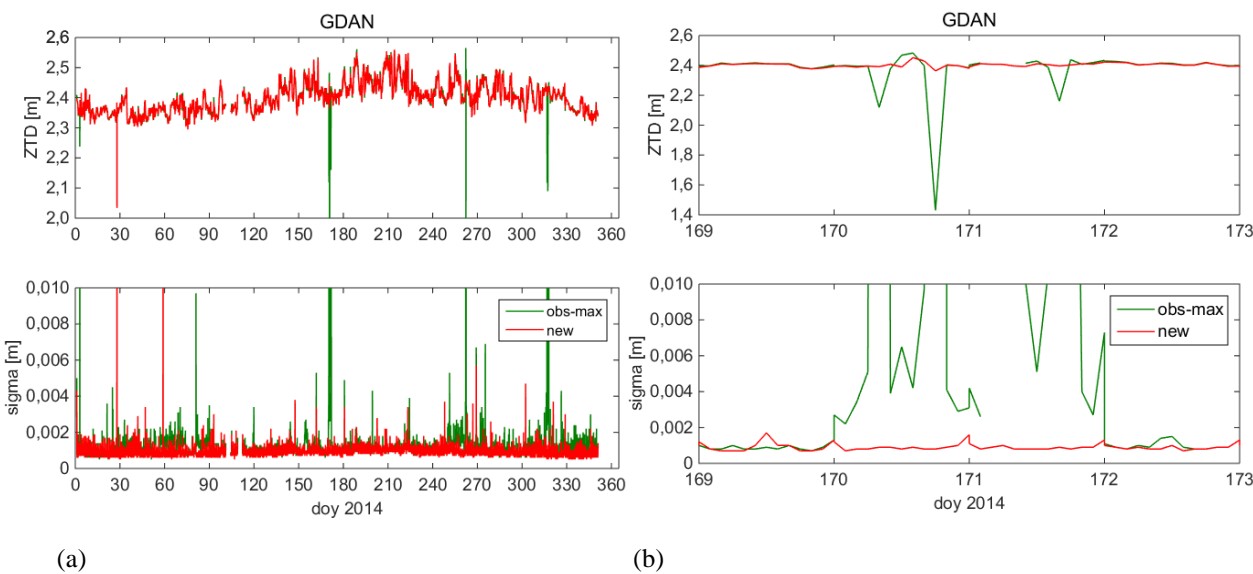

**Figure 10: a) ZTD and formal error of ZTD time series for station GDAN, new (red lines) and obs-max (green lines) baseline strategies. b) Spikes in obs-max solution on days 170-171 are avoided in new solution.**

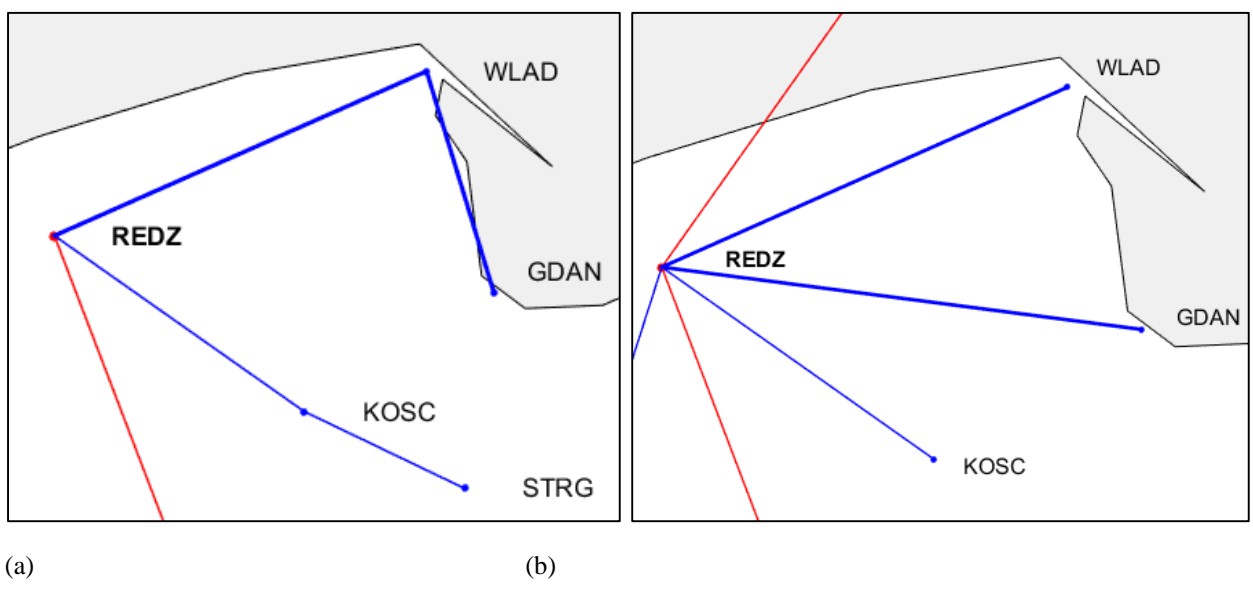

(a)             (b)

**Figure 11: Network design on day 170, a) obs-max solution, b) new solution.**

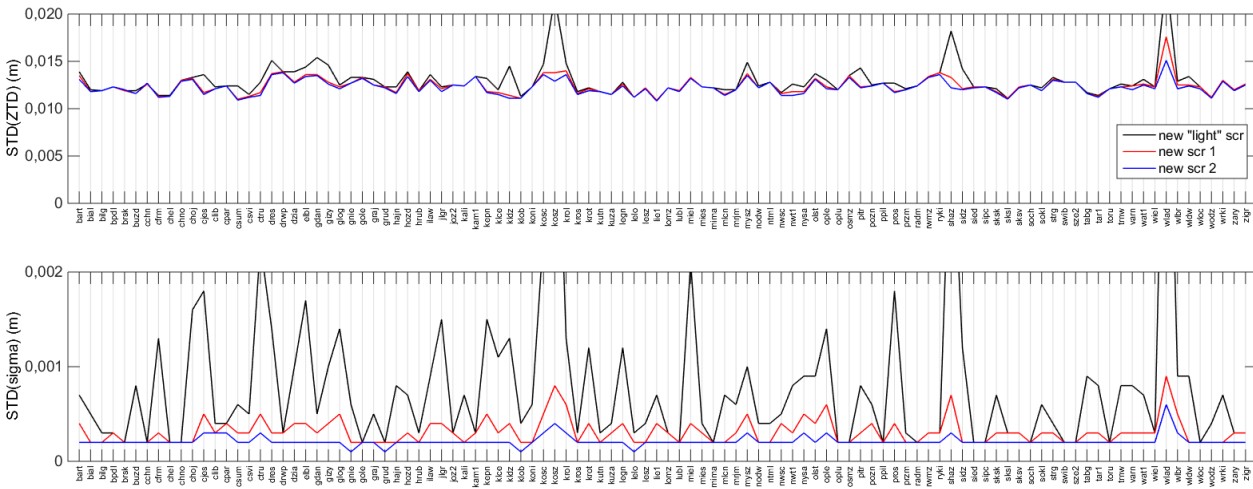

**Figure 12: Distribution of standard deviations of ZTD forward differences and formal error of ZTD at 104 common ASG-EUPOS stations, for the new processing strategy and different screening variants (see text).**

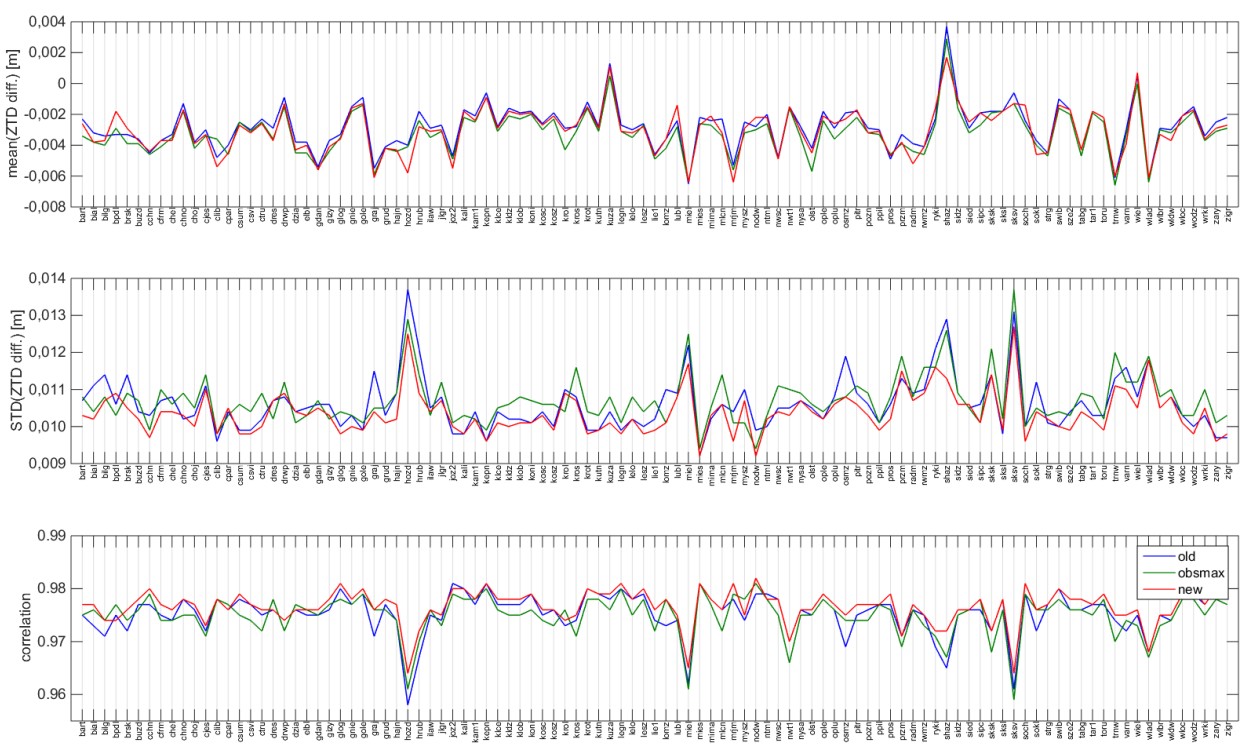

**Figure 13: Distribution of mean differences, standard deviations of differences, and correlation coefficients of ZTD estimates from GPS and ERA-Interim at 103 common ASG-EUPOS stations. GPS data for the three different processing strategies, after screening variant No. 2, are shown.**