# Peer review of "Reduction of ZTD outliers through improved GNSS data processing and screening strategies"

_Atmospheric Measurement Techniques, 2017_

## Referee Comment (RC1) · Anonymous Referee #1 · 15 Jan 2018

Meteorology and climate research benefits from relatively stable Zenith Total Delay (ZTD) time series delivered from regional and continental/global GNSS networks. However, no measurements are perfect and a lot of effort is made for cleaning the data before it can be used. This article is targeted on usage of different GNSS data processing and screening strategies for reduction of deteriorating impact caused by ZTD outliers, particularly due to the gaps in GNSS data time series and temporary reconfiguration of network geometry.

The topic is relevant to the scope of AMT and the results are attractive for any GNSS data analyst working for meteorological or climate applications, including cal-

ibration/validation of different instrumentation.

The techniques and baseline design strategies suggested and demonstrated by the authors are applicable on double differenced strategy of GNSS data processing, particularly for using Bernese v.5.2 software. The post-processing screening strategies used and demonstrated can be considered generic. The impact of data gaps on the number of outliers is properly analysed and improvement of data quality is demonstrated by different pre-configuration of the GNSS network. A new baseline strategy is elaborated and tested, significantly reducing the number of outliers compared to the "old" widely-used strategy in GNSS software for positioning.

Detailed analysis (based on experimental work with data from Polish national network and EPN) is given for each strategy, the efficacy of different strategies is compared between each other and the quality of resulting ZTD time series is compared with data from ERA-Interim reanalysis.

The results are illustrated with appropriate graphs and tables.

The abstract provides a brief overview and a concise summary. The overall structure of the article is clear.

Questions:

1. P.1, Line 17: "... maximizing the number of observations strategy" in many GNSS software ...". The work is done with Bernese v.5.2. Any suggestions for non-Bernese, where the suggested network design could be possible? Is it so, that the automatic network modification (p. 5, line 1), happens with Bernese by default and the new baseline strategy (section 4) has something to do with post-processing, or is it like reconfiguring the Bernese-processing according to the results from initial solution?

2. Initial data processing strategy (section 2): According to Tregoning and Herring (2006) a priori zenith hydrostatic delay errors project into GPS height estimates and errors in zenith delay estimates. Is there any reason why the realistic meteorological

situation is not considered important, or is this effect estimated negligible for initial data processing strategy? Not always can be relied on standard atmosphere model.

Tregoning, P., and T. A. Herring (2006), Impact of a priori zenith hydrostatic delay errors on GPS estimates of station heights and zenith total delays, Geophys. Res. Lett., 33, L23303, doi:10.1029/2006GL027706.

———————————

---

## Referee Comment (RC2) · Anonymous Referee #2 · 19 Jan 2018

The manuscript presented a new base-line strategy and new postprocessing screening procedure to reduce ZTD outliers. The authors did a good job in describing the technical details and validated the methods by comparing with ERA-Interim reanalysis. I have two recommendations to improve the manuscript. (1) It would be useful to discuss why the authors are trying to improving the double-difference processing rather than just using PPP in the introduction, just more elaboration on what has been discussed in the "Conclusions" section. (2) I would recommend that the authors discuss the potential application of proposed methods, such as applying them to historical data to generate a new data with less outliers or to the operational processing to improve future ZTD estimates. So the whole community can benefit.

---

## Author Comment (AC1) · 26 Jan 2018

We would like to thank the Anonymous Referee #1 for valuable comments and suggestions, we are pleased to answer all the questions.

Questions:

1. P.1, Line 17: "...maximizing the number of observations strategy" in many GNSS software...". The work is done with Bernese v.5.2. Any suggestions for non-Bernese, where the suggested network design could be possible? Is it so, that the automatic network modification (p. 5, line 1), happens with Bernese by default and the new

baseline strategy (section 4) has something to do with post-processing, or is it like reconfiguring the Bernese-processing according to the results from initial solution?

Answer: Our strategy can be used in any software in which the user can modify the baseline design. In case of Bernese software, we do initial analysis of the available GPS data from the reference network only to get the number of observations per baseline (we do not carry out any initial float solution, see section 4). Then we apply our strategy to construct optimal baselines, and next we run the complete processing. We hope this clarifies this issue.

2. Initial data processing strategy (section 2): According to Tregoning and Herring (2006) a priori zenith hydrostatic delay errors project into GPS height estimates and errors in zenith delay estimates. Is there any reason why the realistic meteorological situation is not considered important, or is this effect estimated negligible for initial data processing strategy? Not always can be relied on standard atmosphere model.

Tregoning, P., and T. A. Herring (2006), Impact of a priori zenith hydrostatic delay errors on GPS estimates of station heights and zenith total delays, Geophys. Res. Lett., 33, L23303, doi:10.1029/2006GL027706.

Answer: Tregoning and Herring (2006) showed that unrealistic surface pressure used to calculate hydrostatic delay led to errors in estimated ZWD and station heights. Hydrostatic delay errors project into GPS height up to -0.2 mm/hPa, what causes height errors of up to 10 mm and seasonal variations of up to 2 mm amplitude. Errors in ZTD estimates are about half of magnitude of the height errors.

In our processing for all three strategies, a priori meteorological parameters from the Global Pressure and Temperature (GPT) model were used together with Global Mapping Function (GMF) and Chen-Herring gradient model. We realized that this was not specified in the text, hence we will add a following sentence (Page 3; line 8-9):

"In the processing for all three strategies, a priori tropospheric delays were computed

from the Global Pressure and Temperature (GPT) model combined with the Global Mapping Function (GMF) (Boehm et al., 2006) and Chen-Herring gradient model (Chen and Herring, 1997)."

According to Tregoning and Herring (2006), using a priori atmospheric pressure from the GPT model eliminates the majority of the mean height biases caused by a standard/constant pressure value (as it was done in the past).

Note, however, that in this paper we aim at improving the baseline design strategy only, and leave the discussion on a priori hydrostatic delay and its effects for future study.

---

## Author Comment (AC2) · 26 Jan 2018

First, we would like to thank Anonymous Referee #2 for all the comments which helped us to improve the manuscript. Please find below detailed clarifications and responses to your comments.

(1) It would be useful to discuss why the authors are trying to improving the double-difference processing rather than just using PPP in the introduction, just more elaboration on what has been discussed in the "Conclusions" section.

Answer: We would like to include the following paragraph in the introduction (line 3,

page 2):

"Another approach of satellite data processing which can be used to estimate GPS ZTD is the precise point positioning (PPP) technique. Since PPP allows to process each station individually, there is no direct propagation of errors between stations. However, the accuracy of ZTD estimates from PPP processing depends strongly on the quality of satellite orbits and clocks. In our study, we focused on improving the double-difference processing because most EPN and E-GVAP analysis centres rely on a network approach utilizing double-difference observations, and many of them use Bernese GNSS Software v.5.2 (Dach et al., 2015)."

(2) I would recommend that the authors discuss the potential application of proposed methods, such as applying them to historical data to generate a new data with less outliers or to the operational processing to improve future ZTD estimates. So the whole community can benefit.

Answer: Discussion about potential application as already provided in the conclusions (lines 28-31, page 11), nevertheless we will complete the conclusions as suggested by the referee:

"The improved processing strategy may be also an interesting approach for reprocessing historical data to generate a new data with less outliers or to the operational processing to improve future ZTD estimates. More accurate and stable ZTD series may be produced in this mode, and the impact of equipment changes may be more easily detected in the double difference residuals than in zero difference residuals."